# Mindfulness-Based Interventions for the Recovery of Mental Fatigue: A Systematic Review

**DOI:** 10.3390/ijerph19137825

**Published:** 2022-06-26

**Authors:** Shudian Cao, Soh Kim Geok, Samsilah Roslan, Shaowen Qian, He Sun, Soh Kim Lam, Jia Liu

**Affiliations:** 1Faculty of Educational Studies, University Putra Malaysia, Seri Kembangan 43300, Malaysia; caoshudian0516@163.com (S.C.); kims@upm.edu.my (S.K.G.); samsilah@upm.edu.my (S.R.); 2Department of Physical Education, Wuhan Sports University, Wuhan 430000, China; qs.91@163.com; 3School of Physical Education Institute (Main Campus), Zhengzhou University, Zhengzhou 450000, China; 4Faculty of Medicine and Health Sciences, University Putra Malaysia, Seri Kembangan 43300, Malaysia; sklam@upm.edu.my; 5Department of Physical Education, Yuncheng University, Yuncheng 041506, China; liujia1986yuncheng@163.com

**Keywords:** mental fatigue, mindfulness, recovery, psychology, sports

## Abstract

**Background:** There is evidence that mindfulness-based interventions (MBI) can help one to recover from mental fatigue (MF). Although the strength model of self-control explains the processes underlying MF and the model of mindfulness and de-automatization explains how mindfulness training promotes adaptive self-regulation leading to the recovery of MF, a systematic overview detailing the effects of MBI on the recovery of MF is still lacking. Thus, this systematic review aims to discuss the influences of MBI on the recovery of MF. **Methods:** We used five databases, namely, PubMed, Web of Science, EBSCOhost, Scopus, and China National Knowledge Infrastructure (CNKI) for articles published up to 24 September 2021, using a combination of keywords related to MBI and MF. **Results:** Eight articles fulfilled all the eligibility criteria and were included in this review. The MBI directly attenuated MF and positively affected the recovery of psychology (attention, aggression and mind-wandering) and sports performance (handgrip, plank exercise and basketball free throw) under MF. However, the interaction did not reach statistical significance for the plank exercise. Therefore, the experience and duration of mindfulness are necessary factors for the success of MBI. **Conclusions:** mindfulness appears to be most related to a reduction in MF. Future research should focus on improving the methodological rigor of MBI to confirm these results and on identifying facets of mindfulness that are most effective for attenuating MF.

## 1. Introduction

Mental fatigue (MF) is a psychobiological state caused by a prolonged period of demanding cognitive activity. An increasing number of people suffer from MF. For example, since the emergence of the COVID-19 pandemic, peoples’ mental health has been adversely affected. Studies have reported that people associated with the medical profession, such as nurses and recovered and isolated individuals, are likely to experience MF [1,2,3]. This prevalence of MF has implicated a lot of aspects of daily life. MF results in an acute feeling of tiredness and a decreased cognitive ability [4,5,6]. Studies have shown that people exposed to a high depletion manipulation reported feeling more fatigued than those exposed to a low depletion manipulation [7,8,9]. These individuals then exhibited differential levels of self-control. Thus, a previous study proved that perceptions of MF are related to variations in resource depletion [10]. The literature on self-control depletion showed that there are a lot of adverse effects of low self-control resources on individuals. For instance, peoples’ diets were broken more readily [11], there was increased temptation to indulge in alcohol [12], cheat more often [13], individuals fail to manage their emotions [9], and perform poorer on intellectual tasks [8]. On the other hand, individuals low in self-control might be less likely to follow health guidelines [14], and self-control could be considered in intervention strategies to attenuate adolescent problematic mobile phone use, due to the loneliness and escape motivation during the coronavirus outbreak [15]. Finally, athletes exerting self-control in fierce competitions would cause lapses of self-control leading to MF [16]. Moreover, attention regulation is a critical component of self-control, especially in the area of sport and exercise [17]. For instance, to complete a game successfully, players have to ignore the irrelevant information and focus on the relevant targets [18]. A study found that participants in the depleted situation were not good at regulating their impulses and had a worse performance in a dart-throwing task [17].

The strength model of self-control is the most popular model to explain the processes underlying MF. It argues that exerting self-regulation in one domain increases the chance of self-regulation failure and depletes this source to some degree [19,20]. However, there is scientific evidence to suggest that mindfulness training has positive effects on the mechanism of self-regulation. For instance, it has been proven to improve emotion regulation [21] and attention regulation [22,23], which are essential parts of self-regulation [24,25]. Mindfulness is rooted in Buddhism [26], wherein it is defined as the awareness that emerges from paying attention to objects on purpose and without judging the unfolding of experience [27].

The model of mindfulness and de-automatization can explain how the former promotes adaptive self-regulation and desirable health outcomes [28]. It shows that four components of mindfulness (awareness, attention, present focus, and acceptance) could instigate four broad subsequent mental processes, including reduction in automatic inference processing, enhancement of cognitive control, facilitation of metacognitive insight, and prevention of thought suppression and distortion, leading to the improvement of self-regulation. Regarding the mechanisms between mindfulness and MF, studies have demonstrated that mindfulness could improve the anterior cingulate cortex (ACC) [29], respiratory sinus arrhythmia (RSA) [30], and prefrontal cortex [31], leading to the recovery of MF.

A study showed that mindfulness was associated with several types of mindfulness trainings (e.g., different types of yoga or meditation) [32]. In a successful mindfulness mediation, trainers experience current feelings, thoughts, and bodily sensations clearly and plainly without judging or evaluating them and acting on the sensation. In a typical mindfulness exercise, meditators aim to focus their attention on a particular experience and be fully aware of this experience, such as one’s breath and the sensations it evokes in various parts of the body [21,33]. This systematic review defined mindfulness, yoga, or meditation interventions related to mindfulness training as mindfulness-based interventions (MBI).

An increasing number of articles have focused on how to recover from MF. However, there is no systematic review investigating the effects of MBI on the recovery of MF. Hence, this systematic review aims to identify the effect of MBI on the recovery of MF.

## 2. Method

### 2.1. Protocol and Registration

This systematic review was conducted following the PRISMA guidelines [34]. This title has already been registered on the Platform of Registered Systematic Review and Meta-analysis Protocols, and the registration number is INPLASY2021120022.

### 2.2. Eligibility Criteria

PICOS (population, intervention, comparison, outcome and study designs) criteria were used as the inclusion criteria (see Table 1). Studies were required to fulfill the following five inclusion criteria: (1) MF was observed in participants; (2) studies published in English with the application of MBI, regardless of population characteristics, such as age, gender, and ethnicity; (3) randomized controlled trials (RCTs), non-randomized controlled trials (nRCTs) and non-randomized non-controlled trials (nRnCTs) with two or more groups and single-group trials; (4) MBI was used for intervention; (5) comparison was made without MBI and MF-inducing possibility. The exclusion criteria were as follows: (1) systematic review; (2) medical or neuropsychological studies.

### 2.3. Information Sources and Search Strategy

The search was conducted on 24 September 2021, and the following databases were used: PubMed, Web of Science, EBSCOhost (SPORTDiscus and Psychology and Behavioral Sciences Collection), Scopus, and China National Knowledge Infrastructure (CNKI). The search terms were “mental fatigue” OR “cognitive fatigue” OR “mental effort” OR “cognitive effort” OR “mental exertion” OR “ego depletion” AND mindful* OR meditat* OR yoga. In addition, the related reference lists in the included articles were screened. A search was conducted by title, abstract, and subjects in each database. Finally, there was no language-based limitation imposed. The complete search strategy of all the databases is summarized in Table 2.

### 2.4. Study Selection

EndNote software was used to remove the duplicates of the retrieved articles. The results (titles and/or abstracts) of the studies retrieved using the search strategy and the titles and/or abstracts of studies from other sources were independently screened by two review authors (Cao and Qian) to identify studies that may fulfill the above inclusion criteria. The reviewers reviewed these studies according to the standard of PICOS. The two review authors extracted data independently, and the differences were determined and resolved through discussion. It was decided that they would discuss with the third author if required.

### 2.5. Date Extraction

After the completion of the search, data were extracted from articles including (1) title, author, and publication year; (2) research design; (3) Sample size; (4) participant characteristics (gender, age, etc.); (5) intervention features (duration, type); (6) comparison (type); (7) measurement tool; (8) research outcomes. After the information was extracted into standard form, another author checked it.

### 2.6. Quality Assessment

The quantitative assessment tool “QualSyst” was used to assess the methodology quality (Kmet and Lee, 2004). It contained 14 items (see Table 2). The score depended on the degree to which the specific criteria were met (yes = 2, partial = 1, no = 0). Items that did not apply to a particular study design were marked as not applicable, i.e., “NA” and excluded from the total calculation of the summary score. A score of ≥75% indicated strong quality, a score of 55–75% indicated moderate quality, and a score of ≤55% indicated poor quality.

## 3. Result

### 3.1. Study Selection

This study screened 255 articles, of which 146 remained after the exclusion of duplicates. After the titles and abstracts were screened, 78 articles remained, and after reading, another 70 articles were eliminated, ultimately leaving 8 relevant articles in the qualitative synthesis. (Figure 1).

### 3.2. Study Quality Assessment

The quality assessment of these eight selected articles determined that seven articles were of strong quality and one was moderate (Table 3).

### 3.3. General Study Characteristics

The eight articles’ population characteristics were reported based on the following:(1)Sample size. The 8 studies have 476 subjects, ranging from 14 [40] to 110 [36] participants, and the mean sample size was 59.5 participants (SD = 32.2);(2)Gender. Five studies investigated males and females [16,35,36,37,38], but two studies just focused on males [39,40], and one just focused on females [41];(3)Age. The age range of subjects ranged from about 13 to 43 years;(4)Mindfulness training background. Regarding the specialty of this study, the background of mindfulness training is essential. Five articles mentioned that the subjects did not have any experience of mindfulness training [16,36,37,39,40]. One article has an experimental group with four-week mindfulness [38], but participants in the other groups did not have mindfulness experience. Two articles did not report the background of mindfulness training [35,41].

### 3.4. Mental Fatigue-Inducing Interventions and Instruments

#### 3.4.1. Intervention Characteristics

There are a number of MF-inducing interventions reported in the pertinent literature. Two studies used the Stroop task to induce MF [39,40]. Other studies used demanding tasks, such as watching the video [35,36], transcribing a neutral text with conditions [16], and the mental calculation task [37]. Axelsen et al. used the AX continuous performance test (AX-CPT) to induce MF [38]. Coimbra et al. asked participants in the experimental group to play in a volleyball competition to induce MF [41]. Regarding the duration of intervention, most studies ranged from 6 min to 15 min. One article, however, intervened for 90 min [38]. Zhu et al. intervened with soccer players before the game and during the rest of each section, but they did not report the specific time [40].

#### 3.4.2. Outcome and Measurement

A wide range of instruments were observed to measure the effectiveness of reducing MF in the selected articles. First, five articles used subjective manipulation checks. While Yusainy and Lawrence [36] and Stocker et al. [16] used two and three questions, respectively, Axelsen et al. used BRUMS [38]. Shaabani et al. used the ego-depletion manipulation check (EDMC) [39], and Coimbra et al. used the MF visual analog scale (VAS) [41]. On the other hand, behavioral manipulation checks were used in two articles; Stocker et al. recorded the number of repetition characters and errors in the intervention [16], and Friese et al. used d2 performance to evaluate the effects of MF [35]. Wang et al. evaluated the effects of MF by the handgrip time [37]. Finally, Zhu et al. did not report the instrument [40].

Six studies showed that MF impacted participants’ performance (e.g., attention, self-control) [16,35,36,37,38,39], but one study presented no significant change between post- and pretest performance [40], while Coimbra et al. did not measure the MF after the MF-inducing intervention (volleyball competition) [41]. The overview of the MF-inducing intervention can be found in Table 4.

### 3.5. Mindfulness-Based Interventions and Mental Fatigue

#### 3.5.1. Intervention Characteristics

Regarding the intervention characteristics, the following two were mentioned:Type of intervention used. Seven studies had brief MBI [16,35,36,37,38,39,40], but one study used an additional experienced MBI as the second group, and another used brief MBI with carbohydrate (CHO)-electrolyte solution as the experimental group [40]. One study used a long-term MBI [41].Duration of each MBI. The brief MBI ranged from 4 to 15 min, and one study had an additional four-week experienced MBI [38]. One study used a two-week MBI, consisting of fourteen sessions, and each session was approximately 10 min.

#### 3.5.2. Outcome and Measure

##### MBI on Psychological Recovery under MF

Two studies directly used the measurement of MF as an assessment tool, and the results showed that MBI decreased MF [40,41]. One study measured the % no go success rate in the Sustained Attention to Response Task (SART) to test the effects of MBI on mind-wandering, which was recovered after MBI [38]. Friese et al. used the d2 test as an assessment tool, and the test performance was increased after the intervention [35]. Yusainy and Lawrence used the Taylor Competitive Reaction Time (TCRT) task to appraise the effect of MBI on aggression, and aggression in the TCRT task was decreased [36].

##### MBI on Sports Performance under MF

Three studies used sport or exercise performance to assess the effect of MBI on MF, including handgrip, plank exercise, and basketball free-throw shooting performance [16,37,39]. The handgrip time was equal from pre- to post-test, which means the mindfulness practice counteracts the effects of MF. The plank exercise and basketball free-throw shooting performance increased between the experimental and control groups, but the score of the former did not reach statistical significance [16]. The overview of MBI can be found in Table 5.

## 4. Discussion

This systematic review provides a comprehensive overview of the effects of MBI on the recovery of MF. The main findings indicated that MBI could aid in the recovery from MF and the performance impaired by MF (e.g., d2 and free-throw shooting performance). The participants were varied in this review (gender and age). Nevertheless, MBI may be an effective recovery strategy for mentally fatigued individuals.

### 4.1. Mental Fatigue-Inducing Interventions

Shaabani et al. used the Stroop test to induce MF. The Stroop task has been a standard intervention in other studies [42,43]. Zhu et al. also used the Stroop task, but this article showed no significant difference in MF from post- to pretest. In detail, they used the Stroop test to induce MF before the experiment and during three minutes of rest in a soccer game [40]. The intervention was not consistent, and the duration of intervention was less than 10 min. Therefore, we supposed that the discontinuity of the intervention and the short intervention time might be the reason.

On the other hand, Yusainy and Lawrence asked participants to watch a video about an interview in which a series of common one-syllable words appeared at the bottom of the screen for 10 s each. Participants in the experimental group were instructed not to read and look at any word on the screen [36]. If they caught themselves looking at the words, they were required to immediately turn their eyes to the interviewee’s face. This intervention has been proven to have the efficiency to induce MF in studies [44,45]. Friese et al. also asked participants to watch a video, but the participants in the experimental group were asked to suppress all emotions that may arise in response to the video [35]. Emotion suppression has also been used to deplete self-control in another study [20]. Stocker et al. required participants to transcribe a neutral German text from the computer screen on paper for six minutes as fast as they could [16]. The participants in the experimental group had to omit all instances of “e” and “n”, which are the most common letters in German. They had to override their writing habits; a high amount of self-control was needed. Another study also proved that the method was successful in inducing MF. Wang et al. opted for a task based on a mathematical mental calculation to induce MF, for which the participants were not allowed to make a draft to arrive at accurate answers. This task was used to generate ego depletion [37]. Owing to the difficulty associated with mental calculation, individuals quickly had the impulse to draft, and restraining the impulse to draft consumed self-control resources [37]. In addition, Axelsen et al. used the intervention of AX-CPT, which compares the number of correct responses from the initial 15 min of the task to the last 15 min to check whether the AX-CPT reduces the working memory that has been proved as a marker of MF [46]. Many studies have used AX-CPT to induce MF successfully [5,47,48]. Finally, Coimbra et al. asked participants in the experiment group to compete in a game of volleyball to induce MF [41]. Volleyball requires psychomotor and perceptual cognitive skills, such as decision-making, co-ordination, and emotional control [49,50]. It also involves open-skill sports unpredictability. Athletes have to react in continuously changing situations and an externally paced environment, requiring higher cognitive demands in planning and strategy, leading to increased MF [51]. Therefore, although Coimbra et al. did not measure the MF during and after each competition, the MF could be induced by competition successfully.

### 4.2. Mindfulness-Based Interventions and Mental Fatigue

In this review, the effects of MBI on psychological recovery and sports performance under MF are introduced.

#### 4.2.1. MBI on Psychological Recovery under MF

Zhu et al. [40] and Coimbra et al. [41] directly measured the MF between the control and experimental groups after MBI to assess the effect. The former asked participants in one experimental group to receive brief MBI coupled with CHO during the half-time of a soccer game, and participants in another experimental group just injected CHO [40]. The results showed that CHO had a significant adverse effect on MF, but MBI coupled with CHO ingestion could alleviate MF. It has been proved that both relaxing breaths and the acceptance of moods could positively affect the brain, and the biological system that regulates the generation of MF could be affected by acceptance and relaxation that could be improved by MBI [52].

Three potential mechanisms discussed in different studies confirmed the relationship between MBI and MF. First, the anterior cingulate cortex (ACC) could result in the causation of MF [29]. The study proved a stronger subgenus and adjacent ventral ACC activity in the MBI condition and likelihood that MBI could recover MF [53]. In addition, the study by Ditto et al. also proved that MBI could improve RSA [30]. This improvement is related to the variation in heart rate in synchrony with respiration increases in the resting state and decreases in conditions of stress or tension [54], and RSA is also a valid and reliable biomarker of emotional regulation capacity in humans [55]. Hence, the increase in RSA has a close relationship to the resting state, and thus relieves MF. Finally, MBI is beneficial to the prefrontal cortex, modulating brain activities in multiple emotion-processing systems [31]. A previous study found that the increase in salivary cortisol concentration associated with competition in wheelchair basketball athletes was attenuated after an 8-week MBI [56]. Kachanathu et al. proved that using four-week MBI before the competition could attenuate the cortisol concentration in the shooter [57]. All the shreds of evidence indicated that MBI could attenuate competition-related MF.

Axelsen et al. used experienced (four-week mindfulness training) and novice mindfulness (on-the-spot MBI) groups to test the effect between four-week MBI and on-the-spot MBI on SART, which was the assessment tool for MF [38]. The results showed that four-week MBI could reduce MF, but on-the-spot MBI did not reduce the effect of MF on the SART % no go success variable. The SART is a task often used to measure mind-wandering indirectly [58]. They also found that experimental mindfulness group had better performance than other groups in the SART test before and after the intervention, which is identical to other findings [59,60]. The study showed that mindfulness could reduce mind-wandering [59]. One study argued that MBI could improve metacognitive regulation, thereby increasing mind-wandering [61]. This awareness allows individuals to return their ideas to the task at hand [60]. In conclusion, MF reduces attention on the task at hand, and mindfulness helps them focus on the task by providing emotional space, which might explain why the participant had a better performance of the SART task, and there were more negligible effects by MF. Finally, the lack of personal experience with mindfulness exercise is likely be the reason why on-the-spot MBI had no significant effect on the performance of the SART task [36].

Friese et al. used the d2 test as an assessment tool to measure the effect of MBI on MF, and the results showed that a brief MBI reduced the effects of self-control depletion [35]. D2 is a standardized test of attention and concentration [62]. Yusainy and Lawrence used an adapted TCRT task to assess the effects of MBI on aggression [36]. The results showed that participants who were depleted and received MBI had lower levels of blast intensity than those who just were depleted but did not receive MBI, which means MBI has a positive influence in reducing post-depletion aggression. It has been proven that self-control failure can result in various types of aggressive behaviors [44,45,63]. A number of studies researched the effects of mindfulness on attention, and they argued that meditation required the beneficial effects of attention control in a task [22,23]. Regarding the relationship between mindfulness and self-control, studies have explained that mindfulness meditation mainly increases awareness of acute inner experiences [21,64], reducing the adverse effects of self-control depletion [65]. On the other hand, mindfulness meditation might result in the feeling of relaxation [66], which could help people boost their self-control performance [67].

In conclusion, MBI positively influences mentally fatigued peoples’ psychology (e.g., attention and self-control).

#### 4.2.2. MBI on Sports Performance under MF

In this systematic review, three articles studied the effect of MBI on sports or exercised performance under MF [16,37,39].

Wang et al. [37] and Shaabani et al. [39] showed that mindfulness recovered the handgrip and basketball free-throw performance of players after MF, respectively. In normal conditions, the mind can adequately maintain central control and protect against functional impairments resulting from a high-pressure condition, disrupting individuals’ attention [68,69]. However, as brief MBI can regulate emotion [70], it may progress performance in delicate perceptual motor tasks by suppressing irrelevant impulses and keeping attention on the task at hand [71,72]. Therefore, MBI can help people restore self-control, manage distracting stimuli, and maintain attention to relevant information (e.g., basketball).

Stocker et al. showed that plank exercise performance was increased after MBI, but it did not reach statistical significance [16]. Their results are surprising, but we have to carry out an in depth discussion about the reasons. First, participants in this study did not have any experience in mindfulness training. The success of mindfulness training might be decided by the experience of how people are familiar with it. On the other hand, the short duration (just four minutes) may also be a critical reason.

## 5. Limitations

First, one selected study used the intervention of mindfulness combined with CHO, which might have interfered with the results. In addition, just three selected articles mentioned the sample size calculation method [16,39,41]. Excessive or inadequate sample sizes and wrong sample size calculation methods can affect the study’s quality, accuracy, and results. Finally, just three selected articles measured the mindfulness state by mindfulness-related questionnaires [36,39,40]. If the mindfulness state was not measured in the study, it may be not clear whether the effects are attributable to the MBI.

## 6. Conclusions

By reviewing the results of eight published studies, the present systematic review provides evidence that MBI could directly attenuate MF. MBI had a positive effect on other psychological performances (attention, aggression and mind-wandering) and sports performance (plank exercise, basketball free throw and handgrip) under MF, but if people lack the experience of MBI or the duration of the intervention is too short, the mind-wandering and plank exercises would not be affected. In addition, an increasing number of researchers have investigated the effect of MF on sports performance, and some of them have proved the adverse effects, but in the present systematic review, just three studies researched the effects of MBI on the recovery of MF in the sports area. Thus, further studies should pay more attention to this. Furthermore, a previous study argued that men improved more in emotion suppression than women after MBI [73]. In this study, five selected articles mentioned the different genders of the participants, but none investigated whether the MBI has a different effect on MF between genders, which should be investigated in the future. Finally, further studies should put more attention on exploring how the complex mechanism of MF is actually influenced by MBI in the short and long-term. Despite some limitations, the present review presents a synthesis of the existing findings and provides theory and future research on MF and MBI.

## Figures and Tables

**Figure 1 ijerph-19-07825-f001:**
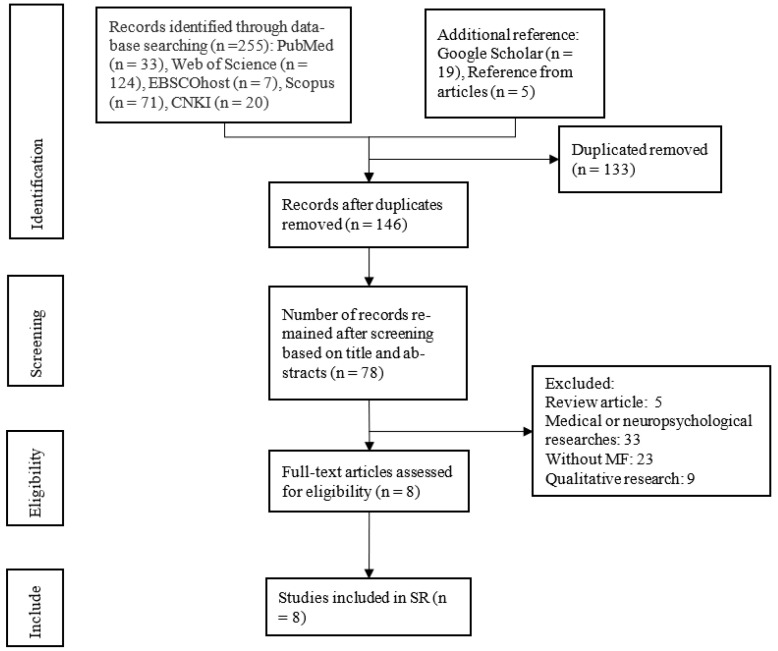
Systematic review search and screening procedure.

**Table 1 ijerph-19-07825-t001:** Inclusion criteria according to the PICOS conditions.

Items	Detailed Inclusion Criteria
Population	Regardless of population characteristics, such as age, gender, and ethnicity
Intervention	MBI
Comparison	Without MBI and MF-inducing possibility
Outcome	Encompassed the effects of MBI on MF
Study designs	RCTs, nRCTs and nRnCTs

**Table 2 ijerph-19-07825-t002:** Number of hits for the complete search strategy for the databases.

Database	Complete Search Strategy	Hits (24 September 2021)
PubMed	(“mental fatigue” (Title/Abstract) OR “cognitive fatigue” (Title/Abstract) OR “mental effort” (Title/Abstract) OR “cognitive effort” (Title/Abstract) OR “mental exertion” (Title/Abstract) OR “ego depletion” (Title/Abstract)) AND (mindful* (Title/Abstract) OR meditat* (Title/Abstract) OR yoga (Title/Abstract))	33
Web of Science	(TS = (“mental fatigue” OR “cognitive fatigue” OR “mental effort” OR “cognitive effort” OR “mental exertion” OR “ego depletion”)) AND TS = (mindful* OR meditat* OR yoga)	124
EBSCOhost	SU (“mental fatigue” OR “cognitive fatigue” OR “mental effort” OR “cognitive effort” OR “mental exertion” OR “ego depletion”) AND SU (mindful* OR meditat* OR yoga)	7
Scopes	(TITLE-ABS-KEY (“mental fatigue” OR “cognitive fatigue” OR “mental effort” OR “cognitive effort” OR “mental exertion” OR “ego depletion”) AND TITLE-ABS-KEY (mindful* OR meditat* OR yoga))	71
CNKI	TKA = (“mental fatigue” + “cognitive fatigue” + “mental effort” + “cognitive effort” + “mental exertion” + “ego depletion”) and TKA = (mindful* + meditat* + yoga)	20

* Combined keywords were included in the screening process.

**Table 3 ijerph-19-07825-t003:** Quality assessment through QualSyst.

Item No.	Friese et al. [35]	Yusainy and Lawrence [36]	Wang et al. [37]	Stocker et al. [16]	Axelsen et al. [38]	Shaabani et al. [39]	Zhu et al. [40]	Coimbra et al. [41]
I	2	2	2	2	2	2	2	2
II	2	2	2	2	2	2	2	2
III	2	2	2	2	2	2	2	2
IV	2	2	2	2	2	2	2	2
V	0	2	2	2	2	2	2	2
VI	0	0	0	2	0	1	0	0
VII	0	0	0	0	1	2	2	2
VIII	2	2	2	2	2	2	2	2
IX	2	2	2	2	2	2	1	2
X	2	1	2	1	2	2	2	2
XI	2	2	2	2	2	2	2	2
XII	0	0	0	0	0	0	0	0
XIII	2	2	2	2	2	2	2	2
XIV	2	2	2	2	2	2	2	2
Rating	Moderate	Strong	Strong	Strong	Strong	Strong	Strong	Strong

*Note. 2* indicates yes, *1* indicates partial, *0* indicates no, *I* question described, *II* appropriate study design, *III* appropriate subject selection, *IV* characteristics described, *V* random allocation, *VI* researcher blinded, *VII* subjects blinded, *VIII* outcomes measure well defined and robust to bias, *IX* sample size appropriate, *X* analytic methods well described, *XI* estimate of variance reported, *XII* controlled for confounding, *XIII* results reported in detail, and *XIV* conclusion supported by results.

**Table 4 ijerph-19-07825-t004:** Overview of mental-fatigue inducing interventions.

Study	Population (P)	Characteristics	Intervention (I)	Duration	Comparison (C)	Methodological Characteristics	Outcome (O)
Friese et al. [35]	40 FM, 26 M	A = 43.27 ± 11.9 years	Watching video in condition (suppressing all emotions)	6.5 min	Watching video naturally	Not reported	d2 performance (attention and concentration) decreases in I vs. C
Yusainy and Lawrence [36]	58 FM, 52 M	A = 19.52 ± 2.03 years	Watching video with condition	6 min	Watching video without condition	RCT	Aggression increases in I vs. C;
Wang et al. [37]	46 FM, 14 M	University students; A = 21.33 ± 1.00 years	Mental calculation task	15 min	N/A	RCT	Handgrip time decreases in posttest vs. pretest
Stocker et al. [16]	18 FM, 16 M	Freshman studying sport science; A = 20.85 ± 1.31 years	Transcribe a neutral text (omit all letters “e” and “n”)	6 min	Transcribe a neutral text conventionally	RCT	Ego-depletion increases in I vs. C; number of repetition characters decrease and errors increase in I vs. C
Axelsen et al. [38]	47 FM, 43 M	G1 = 23 (A = 34.5 ± 10.3 years) G2 = 22 (A = 35.9 ± 12.3 years) G3 = 21 (A = 32.8 ± 9.3 years) G4 = 24 (A = 34.6 ±10.6 years)	AX-CPT	90 min	N/A	RCT	MF increases in posttest vs. pretest; incorrect responses increase in final 15 min vs. first 15 min; % no go success rate decreases in posttest vs. pretest
Shaabani et al. [39]	72 M	Experienced basketball players; urban league; A = 28.6 ± 4.0; H = 193.0 ± 7.5 cm;	Incongruent modified Stroop color-word task	15 min	Congruent modified Stroop color-word task	RCT	Ego-depletion increases in I vs. C; free throw shooting score decreases in I vs. C
Zhu et al. [40]	14 M	Soccer player; A = 24.3 ± 3.7 years; H = 1.74 ± 0.05 cm; W = 68.3 ± 5.1 kg; VO_2max_: 47.0 ± 4.4 mL/kg/min; average training years: 2.5	Stroop	About 3 min × 3 times	N/A	RCT	MF = in post vs. pretest
Coimbra et al. [41]	30 FM	Volleyball athletes; A = U-16, U-14, and U-13	N/A	N/A	N/A	RCT	Not reported

*Note. A* age, *AX-CPT* AX continuous performance test, *C* control, *FM* female, *H* height, *I* intervention, *G* group, *M* male, *MF* mental fatigue, *N/A* not applicable, *RCT* randomized controlled trial, *U* under, *W* weight.

**Table 5 ijerph-19-07825-t005:** Overview of mindfulness-based interventions.

Study	Population (P)	Intervention (I)	Duration	Comparison (C)	Outcome (O)
Friese et al. [35]	40 FM, 26 M	Meditation	5 min	Six connect-the-dots figures	d2 performance (attention and concentration) increases in I vs. C
Yusainy and Lawrence [36]	58 FM, 52 M	Audio instruction	15 min	Two neutral educational excepts and composing words in condition	Aggression in TCRT task decreases in I vs. C
Wang et al. [37]	46 FM, 14 M	Meditation practice	5 min	N/A	Handgrip time = posttest vs. pretest
Stocker et al. [16]	18 FM, 16 M	Audio mindfulness exercise	4 min	Listening to audiobooks	Plank exercise performance increased in I vs. C but did not reach statistical significance
Axelsen et al. [38]	47 FM, 43 M	1. listening guided mindfulness audio 2. M training and listening guided mindfulness audio	1. 12 min 2. 4-week and 12 min	Sitting and relaxing in front of the desktop PC	1. % No go success rate (mind-wandering) = in I vs. C 2. % No go success rate (mind-wandering) increases in I vs. C
Shaabani et al. [39]	72 M	Audio mindfulness training	15 min	Listening to an audio story	State mindfulness increases in I vs. C; free throw shooting score increases in I vs. C
Zhu et al. [40]	14 M	CHO-electrolyte solution and audio MBI	6 min	CHO-electrolyte solution and travelling introduction audio	MF decreases in I vs. C
Coimbra et al. [41]	30 FM	Mindfulness-based mental training	10 min for 14 times in 2-weeks	Not reported	MF decreases in I vs. C

*Note. C* control, *CHO* carbohydrate, *FM* female, *I* intervention, *M* male, *MF* mental fatigue, *TCRT* Taylor Competitive Reaction Time.

## Data Availability

The datasets generated during and/or analyzed during the current study are available from the corresponding author upon reasonable request.

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
