# Peer review of "Mindfulness-Based Interventions for the Recovery of Mental Fatigue: A Systematic Review"

_ijerph, 2022, doi:10.3390/ijerph19137825_

Round 1
Reviewer 1 Report
--Abstract: sentence 3—what this project does—seems to say the same thing as sentence 2—what has already been done. What is new here?
--The inclusion criteria—keywords MBI (mindfulness-based interventions, including not only RCTs, but also nRCTs and nRnCTs) and MF (mental fatigue)—does little to assess quality of studies included. According to QualSyst reporting, researchers were only blinded in one of the studies, and subjects were only blinded in three studies; not a single study was controlled for confounding! Only three of the studies addressed sports, and only 2 of the 3 claimed statistically significant benefits. Only three studies noted the sample size calculation method. The unevenness/unclear quality of studies reviewed, combined with the small sample size (eight studies) and heterogeneous characteristics of the MBIs and measures of MF, compound to make it unclear that this review warrants generalized conclusions.
--The Conclusion seems to far outstrip the results reported. Does the publication of eight studies of uncertain quality and heterogenous interventions and measures claiming benefits for MF really support the claim that MBI is “effective” and “affordable” for people “everywhere”? Or that it is “especially” helpful as a response to COVID (esp. given that half the included studies were published before 2020)? These claims seem unwarranted.
--The logic upon which the study is premised and the paper is organized seems questionable. A) COVID worsens MF, which weakens self-regulation. B) MBI helps self-regulation. C) There are no systematic reviews of MBI on recovery of MF. D) This systematic review is needed. There are many other ways of addressing self-regulation and/or MF and/or COVID. What is unique about MBI as a response to COVID-related MF that justifies this review and/or its wide-reaching conclusions?
--The authors seem to accept findings they like and to dismiss the relevance of findings they do not like, e.g. the MBI must not have been long enough or the participant must not have had enough personal experience with mindfulness if benefits are not reported.
--The quality of English/grammar/writing needs improvement.
Author Response
Thank you very much for the comments. Please see the attachment about the response.

Reviewer 2 Report
Tables need to be smaller throughout review. Alignment was off throughout the document.
Author Response

(The authors gave the same response as above.)

Reviewer 3 Report
I thank authors for their contribution. The paper presents a typical systematic review paper on the subject of mindfulness-based interventions for the recovery of mental fatigue. A systematic review of this kind could be very useful for researcher or general public as a quick introduction to the topic. While I support this kind of publications I have some additional comments:
1. authors should clearly identify a time frame in the identification phase of the data collection.
2. I would exclude structural elements from the abstract (words - background, method and conclusion)
3. References in tables should not be presented in brackets.
4. Authors starts with the reference to the COVID influence to the mental health and finish conclusion with the reference to the same topic, while the results are not directly related to the pandemic. I would not start the paper with that.
Thank you again and good luck with the publishing!
Author Response

(The authors gave the same response as above.)

Reviewer 4 Report
This manuscript presents a well-written but relatively modest systematic review regarding the effect of Mindfulness Based Interventions (MBIs) in reducing induced mental fatigue. The review was registered, followed clear eligibility criteria and the search was transparent and clearly described. A total of eight articles fit the eligibility criteria and were described, and results interpreted in relation to the main research question. The authors state that overall, the included studies suggest that a brief mindfulness/meditation intervention reduces the effects of acute induced mental fatigue. The manuscript addresses a relevant topic, and is novel as no systematic reviews have yet been performed on this specific topic. There are some weaknesses that should be addressed, listed below.
#1 Though there is no strict threshold regarding the minimum number of studies that should be included in a systematic review, the included number of studies (n=8) is relatively low, especially noting the lack of congruency regarding the exact intervention and index of mental fatigue. I feel, that more nuance should be applied in the conclusion and the preliminary nature of this first systematic review underscored.
#2 In a related vein, most studies that were included did not incorporate an optimal control group to which the effects of mindfulness/meditation were compared. For those studies, this makes it difficult if not impossible to attribute effects to the mindfulness inducing component of the active (MBI) group. This issue deserves some more attention. Where there any checks regarding whether state mindfulness was actually improved following the interventions? That would make it clearer whether effects are attributable to increases of mindfulness.
#3 The differences and overlap between the interventions (mindfulness and meditation) could be addressed more, and perhaps the plural “interventions” should be used in the title.
#4 It is concluded that: “MBI is an effective and affordable recovery strategy for people everywhere and can help individuals with problems related to mental health such as anxiety, nervousness, and depression because of the loss of self-control, especially during the pandemic COVID-19.”. I think this conclusion is too strongly put and not really supported by the presented data. More nuance should be applied. Specifically, all included studies report on the effect of mindfulness/meditation on acute induced mental fatigue, not on the effects on more chronic mental fatigue (and associated mood related symptomatology) induced by long(er) term stressors such as a pandemic.
Author Response

(The authors gave the same response as above.)

Round 2
Reviewer 1 Report
The authors made very modest revisions that did not substantially addressed the concerns articulated in my original review, nor did the cover letter do much to address them.
Author Response
Dear reviewer, I carefully revised the manuscript again according to your advice, and I also explain the points that you mentioned one by one. Please see the attachment. Thank you very much.
I have seen that the editor's decision on the website said my manuscript includes not only RCTs, but also nRCTs and nRnCTs. But actually, the seven of eight selected articles are RCT, just one article did not report the type of design. So I want to explain that the quality of the study was not affected.

Reviewer 4 Report
My comments and suggestions are briefly but sufficiently addressed, and I have no further comments regarding the revised manuscript.
Author Response
Thank you very much for your comments and advice. You gave a relatively high score for my study but you would not like to sign my review report. If you have no further comments on my revised manuscript, can I ask you to sign my review report? I really appreciate you.